# Gender-Specific Risk Factors of Physical Activity-Related Injuries among Middle School Students in Southern China

**DOI:** 10.3390/ijerph16132359

**Published:** 2019-07-03

**Authors:** Dongchun Tang, Weicong Cai, Wenda Yang, Yang Gao, Liping Li

**Affiliations:** 1Injury Prevention Research Center, Shantou University Medical College, Shantou 515041, China; 2Department of Sport and Physical Education, Hong Kong Baptist University, Kowloon Tong, Hong Kong

**Keywords:** physical activity, sports injury, risk factors, adolescents, middle school

## Abstract

This cross-sectional study was carried out to explore the potential risk factors of physical activity-related injuries (PARI) among middle-school students of different genders. Selected by the random cluster sampling method, students aged from 12 to 16 years old in grades 7–8 from six middle schools in Shantou, southern China, were recruited for this investigation in November 2017. Information about socio-demographics, physical activity (PA) exposure time, individual exercise behaviors, risk-taking behaviors, and PARI experiences in the past 12 months was collected. A multivariate logistic regression model was used to explore the risk factors of PARI. A total of 1270 students completed valid questionnaires, with an overall injury incidence of 33.6% (boys: 42.0%; girls: 25.0%), an injury risk of 0.68 injuries/student/year, and an injury rate of 1.43 injuries per 1000 PA exposure hours. For boys, living in a school dormitory, participating in sports teams, exercising on a wet floor, rebellious behavior, and having longer PA exposure time were the risk factors of PARI. For girls, those who were sports team members, whose parents were divorced or separated, and those with longer PA exposure time were more vulnerable to suffer from PARI. In conclusion, PARI was a health problem among middle school students in southern China. Boys and girls differed in PARI occurrence and were affected by different risk factors, which provides a basis for targeted gender-specific intervention programs to reduce the occurrence of PARI among middle-school students.

## 1. Introduction

Physical activity (PA) is the most active, effective and economic way to improve individual physical fitness and maintain psychological health, especially in the growth and development period in children and adolescents [1,2]. However, the adverse effects of PA participation have also been documented in numerous studies—the increased risk of physical activity-related injuries (PARI) [3,4]. Previous studies revealed that PARI could place a great burden on children and adolescents, resulting in absence from classes and normal PA engagement, physical discomfort, medical consultation and hospitalization [5,6,7].

Compared with adults, the physical and mental development of middle-school students is not yet complete [8]. Furthermore, middle-school students have the most active enthusiasm for PA in their lives and more free time to participate in various kinds of PA, as they experience less pressure in school courses compared with high-school students, which may increase their exposure to PARI [9]. According to the published research, over a third of (37.9%) children and adolescents sustained PARI, with a higher injury incidence in boys (42.1%) than girls (33.9%) [6].

Previous studies have mainly looked at the nature of PARI [3,4,5]; only a few focused on the potential risk factors of PARI among middle school students [10]. An accepted methodology reported that injury prevention measures include four steps [11]: (1) identify and describe the severity; (2) determine the risk factors and mechanisms; (3) introduce measures which are likely to reduce the injury risk in the future; (4) repeat the first step to evaluate the effectiveness of the measures. The risk factors and mechanisms identified in the second step are the basis for the implementation of injury-prevention strategies. Several subject-related factors such as age, gender, body mass index (BMI), grade, PA level, exercise behaviors, and family context were associated with PARI [6,10,12]. In addition, environmental factors like weather, playgrounds and facilities of PA also influence the occurrence of PARI [13]. However, few studies have addressed the relationship between individual risks and PARI outcome, and the differences in risk factors for PARI between boys and girls are poorly understood. 

Consequently, we carried out this cross-sectional study to explore the potential risk factors for PARI occurrence among middle-school students of different genders, aiming to provide a basis for targeted gender-specific prevention strategies to reduce the occurrence of PARI.

## 2. Materials and Methods 

### 2.1. Study Sample

We selected eligible schools by the method of random cluster sampling, on the basis of the administrative areas and locations in Shantou. Students in grades 7 and 8 (aged from 12 to 16 years old) from six middle schools were recruited to take part in this cross-sectional study in 6–26 November 2017. The inclusion criteria were as follows: (a) ability to engage in PA; (b) ability to provide informed parental consent (signed by their parents) for participation in this study. 

### 2.2. Data Collection

Structured self-administered questionnaires with an overall Cronbach’s coefficient alpha of 0.818 were given to all consenting students by our trained personnel, consisting of questions regarding socio-demographics, PA exposure time, individual exercise behaviors, risk-taking behaviors, and PARI occurrence in the past 12 months.

Socio-demographic variables (Table 1) included gender, age, grade, living in a school dormitory (yes or no), being an only child (yes or no), weight and height (BMIs were calculated by weight (kg)/height^2^ (m)), nearsightedness (yes or no), being a sports team member (yes or no), weight loss (yes or no), parental marital status (married, divorced/separated or other), sleep duration, study duration, screen time (including telephone and computer usage), school type (public or private) and location (urban or suburban). BMI categories were based on growth charts produced by the Centers for Disease Control and Prevention, and the students were categorized as overweight and at risk for overweight (BMI ≥ 85th percentile), normal weight (5th to 84th percentile) or underweight (<5th percentile) [14].

The Chinese version of the Children’s Leisure Activities Study Survey (CLASS-C) was used to evaluate participants’ habitual involvement in PA per typical week in the past 12 months, which showed good reliability in our study (Cronbach’s alpha = 0.830) and sound 1-week test–retest reliability in the earlier study [15]. The CLASS-C collects the frequency (total cumulative times) and duration (average minutes each time) of various types of PAs such as running, walking, cycling, hiking, and playing basketball, on both a typical weekday and a typical weekend, respectively. PA volume (total cumulative minutes per week) was calculated by adding the total cumulative minutes on both a typical weekday and a typical weekend, and then the daily minutes of PA participation on average was calculated by PA volume divided by seven. The total yearly exposure time (in hours) of PA involvement was calculated as the weekly PA volume multiplied by 52 weeks and divided by 60 min.

Students were requested to complete 17 questionnaire items regarding individual behaviors when engaging in PA in the past 12 months (Table 2, Cronbach’s alpha = 0.830). For instance, (a) “Did you do warm-up exercises before PA participation?”, (b) “Did you do cool-down exercises after PA participation?”, (c) “Did you use sunscreen during PA participation?”, (d) “Did you drink water regularly during PA participation?”, (e) “Did you wear suitable shoes during PA participation?”, (f) “Did you wear suitable clothing during PA participation?”, (g) “Did you wear protective equipment during PA participation?”, (h) “Did you undertake PA on a wet floor?”, (i) “Did you undertake PA on an uneven floor?”, (j) “Did you undertake PA in insufficient light?”, (k) “Did you undertake PA in extreme hot hours?”, (l) “Did you undertake PA in extreme cold hours?”, (m) “Did you undertake PA with illness?”, (n) “Did you wear glasses during PA participation?”, (o) “Did you wear accessories (i.e., watch, necklace, key, etc.) during PA participation?”, (p) “Did you look over the surroundings before PA participation?”, (q) “Did you pay attention to your physical status like pulse and heartbeat during PA participation?”. For each item, students were asked to choose one of these four options: never, seldom, sometimes, and often.

The risk-taking behaviors of students were assessed by the Chinese version of Adolescent Risk-taking Questionnaire–Risk Behavior Scale (ARQ-RB), which was validated to possess sound reliability in our study (Cronbach’s alpha = 0.772) and have good test–retest reliability in the earlier studies (Cronbach’s alpha = 0.734) [16]. This is a 17-item scale, and each item is rated on a Likert-5 scale ranging from “would never do” (1 point), “would hardly ever do” (2 points), “would do sometimes” (3 points), “would do often” (4 points) to “would do very often” (5 points). The 17 items were divided into four factors of risk-taking behaviors: thrill-seeking (five items, including snow skiing, taekwondo fighting, inline skating, parachuting, and entering a competition), rebellious (six items, including leaving school, underage drinking, smoking, getting drunk, staying out late, and drinking and driving), reckless (two items, including taking drugs and having unprotected sex), and anti-social behavior (four items, including overeating, teasing and picking on people, cheating, and talking to strangers) [16]. The higher the score for each risk-taking behavior factor, the stronger the desire to engage in this certain type of risk behavior.

PARI experiences in the past 12 months were also collected. The definition of PARI described by Bloemers F. et al. was applicable to this study [17]. An acceptable PARI must meet at least one of the following criteria: the student (a) has to stop the current PA and/or; (b) cannot participate in the next planned PA and/or; (c) cannot go to the class next day and/or; (d) has to seek medical treatment (including first-aid, seeing a doctor or receiving physical therapy, but excluding those using bandages only) [17]. All participants were asked to report and count their PARI episodes according to the above four criteria. In addition, injured students were required to provide details of each PARI episode including time, place, cause, mechanism, type, injured body part, PA involved in injury, treatment, etc. All the detailed information helped to validate the measure of outcome PARI.

### 2.3. Procedures

Prior to the survey, an explanatory statement concerning the study and consent forms were required to be signed by students’ parents. These documents were given to 1320 potential students, with a response rate of 97.6% (1288). The self-reported questionnaires were distributed subsequently to all consenting students during school hours. The purpose of this study was explained to the participants before they filled out the questionnaires, and our trained investigators would answer any questions that arose by the subjects during the session. Finally, 18 students were excluded due to their more than 80.0% incompletion rate, leading to a final valid sample size of 1270 with an effective completion rate of 98.6%.

This study was conducted strictly in accordance with the Declaration of Helsinki, and was approved by the Shantou University Medical College Ethics Committee (SUMC-2018-44).

### 2.4. Statistical Analysis

Categorical and continuous variables were presented as number (percentage) and mean and standard deviation (SD), and tested by Pearson’s chi-square tests and independent-sample *t* tests for differences between the PARI and non-PARI groups, respectively. The injury risk was calculated by dividing the total amount of injuries by the number of students, and the injury rate was calculated as the total number of injuries per 1000 PA exposure hours. Poisson distribution was used to calculate the 95% confidence intervals (CIs) of injury risk and injury rate [18]. All significant variables tested by chi-square or *t* tests were included in the multivariate logistic regression model in order to explore the potential risk factors of PARI. Statistical analyses were performed using SPSS 23.0 software (SPSS Inc., Chicago, Illinois, USA). A two-tailed *p*-value of less than 0.05 was considered statistically significant.

## 3. Results

### 3.1. Socio-Demographic Characteristics

A total of 1270 eligible students participated in this study, with a mean age of 13.12 (SD: 0.83). Among the whole sample, 427 students (33.6%) sustained at least one PARI episode in the past 12 months, and there was a significant difference (*χ*^2^ = 40.870, *p* < 0.001) in injury incidence between boys (42.0%, 270/643) and girls (25.0%, 157/627). In total, 863 PARI episodes were reported by all injured students (boys: 575; girls: 288), which equals an overall injury risk of 0.68 injuries/student/year (95% CI: 0.64–0.73) and boys had a significantly higher injury risk (0.89, 95% CI: 0.82–0.97) than girls (0.46, 95% CI: 0.41–0.52) (*p* < 0.05). Significant differences were found in daily PA exposure time between genders (boys: 97.25 min/day; girls: 59.31 min/day; *t* = 7.925, *p* < 0.001). The yearly PA exposure time of the whole study sample was calculated as 604962.80 h (boys: 379350.40; girls: 225612.40). The overall injury rate was thus calculated as 1.43 injuries per 1000 PA exposure hours (95% CI: 1.34–1.53), with a higher injury rate among boys (1.52, 95% CI: 1.40–1.65) than girls (1.28, 95% CI: 1.14–1.44).

For boys, those studying in private schools, living in a school dormitory, participating in a sports team, and having longer screen time and PA exposure time were more likely to suffer from PARI (all *p* < 0.05). For girls, those who were trying to lose weight, participated in a sports team, had longer screen time and PA exposure time, and whose parents were divorced or separated were more vulnerable to sustain PARI (all *p* < 0.05) (Table 1).

### 3.2. Physical Activity (PA)-Related Behaviors

As shown in Table 2, those undertaking PA frequently in extreme hot or cold weather were more vulnerable to suffer from PARI, for both boys and girls (all *p* < 0.05). Moreover, boys exercising frequently on a wet or uneven floor were more likely to sustain PARI, whereas the PARI occurrence of girls was associated with using sunscreen, and wearing suitable clothing and glasses when participating in PA (all *p* < 0.05).

### 3.3. Risk-Taking Behaviors

As presented in Table 3, the PARI group had a significantly higher total score of risk-taking behaviors than the non-PARI group, for both boys and girls (both *p* < 0.05). In particular, for boys, injured students scored significantly higher in thrill-seeking, rebellious, and anti-social behaviors (all *p* < 0.05), while for girls, the PARI group only scored higher in thrill-seeking behavior (*p* < 0.05). 

### 3.4. Multivariate Logistic Regression of Boys

Based on the significant variables identified by a chi-square or *t* test, a multivariate logistic regression model was performed to estimate odds ratios (ORs) and corresponding 95% CIs for PARI among boys, and the results are displayed in Table 4. Compared with their counterparts, sports team members had a greater risk of sustaining PARI (OR = 1.487, 95% CI: 1.042–2.121). Exercising on a wet floor elevated the possibility of PARI, and a dose-response effect could be observed (OR: 1.847–3.013). In addition, those who lived in a school dormitory, had a higher score in rebellious behavior, or a longer PA exposure time were also more likely to suffer from PARI (ORs = 1.858, 1.116 and 1.007, respectively).

### 3.5. Multivariate Logistic Regression of Girls

Similarly, the multivariate logistic regression model was used to estimate the potential risk factors of PARI for girls, whose results in the final model are shown in Table 5. Participating in sports team increased the odds of suffering from PARI (OR = 2.471, 95% CI: 1.662–3.673). Moreover, those whose parents were divorced or separated, or had a longer PA exposure time were more likely to experience PARI (ORs = 2.962 and 1.013, respectively).

## 4. Discussion

This cross-sectional study found that approximately one-third (33.6%) of middle-school students in southern China sustained at least one PARI in the past 12 months, which was higher than in our previous study conducted in 2015 (25.1%) [10]. This percentage also differed from some results reported elsewhere (47.0–57.4%) [19,20,21], but was similar to other findings (32.2% and 37.9%) [4,6]. In addition, our study revealed an overall injury risk and injury rate of 0.68 injuries/student/year and 1.43 injuries per 1000 PA exposure hours, which were different from those observed in other reports [17,22]. These differences could be explained by the different study designs [21,23]. Additionally, the range of PA modalities varied from other research, which was associated with the occurrence of PARI [3,4]. The definition of PARI might be attributable to this difference as well. In contrast to previous studies [19,20,21], we took into consideration injuries leading to class absence. Moreover, different age groups, time periods and regions in data collection might also contribute partly to this disparity [24,25]. Collectively, we should pay adequate attention to the PARI problem among middle-school students, particularly when promoting a physically active lifestyle.

It is worth noting that boys had a significantly higher PARI incidence, injury risk and injury rate than girls (42.0% vs. 25.0%, 0.89 vs. 0.46 and 1.52 vs. 1.28) and different potential risk factors were found between genders, which was consistent with previous research [5,17]. Compared with girls, boys tend to be more motivated and active to engage in PA [26,27], which might elevate their exposure to PARI. This was supported by our results (boys: 97.25 min/day; girls: 59.31 min/day). Besides, boys seem to be more likely to engage in competitive activities like basketball and football. These activities involve a high rate of contact, sprinting, jumping and/or pivoting, which are major injury mechanisms [28]. The great intrinsic motivation and self-determination of boys may play a part as well [29]. These gender-specific characteristics might lead to deviation in the analysis of risk factors. If a risk factor for PARI has different effects on boys and girls, it may be difficult to reveal the real association when analyzing the combined data. Therefore, in order to avoid this problem, we explored the potential contributors to PARI occurrence for boys and girls, respectively. 

In this study, we found that boys living in a school dormitory were more likely to suffer from PARI compared with non-boarders (OR = 1.858). Several reasons might be attributed to this disparity. First of all, according to the results of this study, students living in a school dormitory engaged in more study (boarders: 9.46 h/day; non-boarders: 7.95 h/day) and screen time (boarders: 2.82 h/day; non-boarders: 2.68 h/day), contributing to a significantly long duration of sedentary activity. This would limit boarders from PA participation, which was supported by our data (boarders: 59.90 min/day; non-boarders: 73.34 min/day). The longer sedentary duration and higher physical inactivity levels were harmful for their individual physical well-being [30]. Poor physical condition had a negative impact on the performance of adolescents’ PA, and led to an increased likelihood of suffering from PARI. Moreover, compared with non-boarders, boarders are independent of parental supervision, which may lead them to take part in more high-risk activities, and thus expose themselves to more injuries during PA participation [31]. Furthermore, nutrient intake was one of the key factors affecting the occurrence of PARI [32]. Students living in a school dormitory had more chances to take in junk food because they are away from the control of their parents, which may lead them to have an unhealthy body and result in a higher risk of PARI [33]. This implies that schools and parents should concentrate on the cultivation of students’ good eating habits. However, living in a school dormitory did not affect the occurrence of PARI for girls, as they had higher levels of self-control [34]. Therefore, more supervision should be warranted, especially for boys who are boarders, to effectively reduce the occurrence of PARI.

Despite the fact that the Chinese government has promoted a physically active lifestyle among children and adolescents in recent decades, our study revealed that only approximately half (50.4%) of middle school students engaged in PA daily for more than 60 min. Long screen watching time and homework time were attributable to low PA engagement [35]. In this study, we found that PA exposure time obtained significant differences between genders, and similar results were reported by Simona et al. [36]. In addition, the significant relation between PA involvement and PARI occurrence was found in both boys and girls—longer PA exposure time was a risk factor of PARI. In line with previous studies, a higher frequency, longer duration, and higher intensity of PA participation were associated with an increased risk of PARI [37,38]. This is contrary to the initial intention of promoting PA participation to maintain physical well-being, as PARI can lead to a great burden on children and adolescents [7]. PARI can result in PA absenteeism or medical attention, which may reduce students’ enthusiasm for PA participation and even prevent them from exercising [39]. Therefore, injury-prevention strategies are urgent to reduce PARI occurrence when promoting PA participation. 

Consistent with results reported elsewhere [4], our study found that subjects participating in sports teams were more likely to suffer from PARI in comparison with their counterparts among both boys (OR = 1.487) and girls (OR = 2.471). Sports team members generally focus on one certain type of PA, which may easily lead to overuse injuries of a specific body part [40]. Additionally, sports team members spent a lot of time in training to improve their sports performance [4], and in this way increase their intensity and duration of PA and further lead to more exposure to PARI (97.25 min/day vs. 68.69 min/day in our study). Moreover, the fierce confrontation in the process of sports competition might also play a role [41]. Future injury-prevention measures should focus on this vulnerable population.

Although exercising in outdoor environments greatly benefits individual health, uncertain factors like playground and weather conditions may influence the risk of suffering from PARI [13,42]. In line with earlier studies [43], our study revealed that exercising on a wet floor frequently significantly elevates the likelihood of sustaining PARI for boys. It is easier to slip and fall down when undertaking PA on a wet floor, especially when moving at a high speed or engaging in vigorous intensity outdoor activities [43,44]. Moreover, this might be related to their poor individual safety awareness as well [10]. Furthermore, as a major determinant of injury, we found that individual risk-taking behaviors were also associated with the occurrence of PARI. Boys with high-risk rebellious behavior were at a higher risk of PARI (OR = 1.116). Except for the poor safety consciousness, individual cognitive performance, impulsiveness, and adventurousness also make a contribution [29,45], which might cause individuals to distort the potential risk of a specific behavior. The optimization of environments can lead to an amelioration of individual risks for injury and may protect individuals from the occurrence of PARI associated with risk-taking behaviors [46], and this can serve as a risk reduction strategy with regard to PARI occurrence. Collectively, there is an urgent call for multifaceted measures to provide safe and supportive environments, reduce individual unsafe exercise and risk-taking behaviors, and improve safety awareness when promoting students to take part in PA actively.

Family context was also identified as a potential contributor to injury [47]. Consistent with a previous study [10], we found that female students living in families where the parents are divorced or separated had more possibility to suffer from PARI. This might be due to the fact that children living with single parents might have an increased risk of psychological, emotional, and behavioral problems [48,49]. In addition, a lack of parental concern might also be related to the occurrence of PARI [31]. Thus, children in single-parent families are a high-risk, vulnerable group, and future studies should place great emphasis on this population.

The valuable findings of this study arose from the fact that multidimensional factors were included to explore the risk factors of PARI occurrence for different genders. However, some potential limitations remain to be noted. Firstly, the nature of a cross-sectional design limits us from being able to derive causal and temporal inferences between PARI occurrences, and the potential risk factors in our study should be viewed as exploratory due to the absence of longitudinal data. For example, it is plausible that injured students would rate their level of PA higher if they were not injured, due to the adverse influence of injury on PA level. Likewise, it is also plausible that those students who take part in PA more often might remember their PARI experience more clearly, since their injuries might have a bigger impact on these physically active students. However, the rationale between PA levels and PARI occurrence could not be explained, since we captured the PA exposure time only once in this study. Our team is carrying out a prospective design to confirm the results obtained in the present study. Moreover, the data in this study may be affected by recall and reporting bias because it was collected through a self-administered questionnaire. For instance, students might over-report their height and under-report their weight, which would affect the categories of BMI. Finally, our study did not analyze overuse and acute injuries separately. This may lead to deviation in the association between the potential risk factors and PARI, since the mechanisms behind the two types of injuries are different. Therefore, further study should take the above limitations into account. 

## 5. Conclusions

In conclusion, PARI was a health problem among middle-school students in southern China, and PARI occurrence and its risk factors varied between different genders. For boys, living in a school dormitory, participating in sports teams, exercising on a wet floor, rebellious behavior, and having longer PA exposure time were the risk factors of PARI occurrence. For girls, those who were sports team members, whose parents were divorced or separated, and those with longer PA exposure time were more vulnerable to suffer from PARI. These findings provide a good basis for gender-specific prevention programs to reduce the occurrences of PARI among middle-school students.

## Figures and Tables

**Table 1 ijerph-16-02359-t001:** Comparison of socio-demographics in physical activity-related injuries (PARI) and non-PARI groups among different genders.

Characteristics	Boys (*n* = 643)			Girls (*n* = 627)		
	PARI (*n* = 270) *n* (%)	Non-PARI (*n* = 373) *n* (%)	*χ*^2^/*t* *	PARI (*n* = 157) *n* (%)	Non-PARI (*n* = 470) *n* (%)	*χ*^2^/*t* *
Grade			0.045			2.387
7th	142 (42.4)	193 (57.6)		85 (27.8)	221 (72.2)	
8th	128 (41.6)	280 (58.4)		72 (22.4)	249 (77.6)	
Type of school			5.191 ^1^			0.249
Public	155 (38.6)	247 (61.4)		104 (25.7)	301 (74.3)	
Private	115 (47.7)	126 (52.3)		53 (23.9)	169 (76.1)	
Location of school			0.397			0.074
Urban	94 (43.7)	121 (56.3)		57 (25.7)	165 (74.3)	
Suburban	176 (41.1)	252 (58.9)		100 (24.7)	305 (75.3)	
Living in a school dormitory			7.541 ^2^			0.094
Yes	120 (48.8)	126 (51.2)		53 (24.3)	165 (75.7)	
No	150 (37.8)	247 (62.2)		104 (25.4)	305 (74.6)	
Only child			0.700			1.033
Yes	51 (45.5)	61 (54.5)		21 (30.0)	49 (70.0)	
No	219 (41.2)	312 (58.8)		136 (24.4)	421 (75.6)	
Body mass index (BMI) (kg/m^2^)			0.577			0.203
Underweight	41 (39.8)	62 (60.2)		7 (21.9)	25 (78.1)	
Normal weight	217 (42.7)	291 (57.3)		126 (25.4)	370 (74.6)	
Overweight ^4^	12 (37.5)	20 (62.5)		24 (25.5)	70 (74.5)	
Nearsightedness			1.947			2.305
Yes	129 (39.3)	199 (60.7)		82 (22.8)	278 (77.2)	
No	141 (44.8)	174 (55.2)		75 (28.1)	192 (71.9)	
Weight loss			0.732			6.304 ^1^
Yes	45 (45.9)	53 (54.1)		45 (33.3)	90 (66.7)	
No	225 (41.3)	320 (58.7)		112 (22.8)	380 (77.2)	
Sports team member			16.421 ^3^			21.716 ^3^
Yes	120 (52.6)	108 (47.4)		77 (36.3)	135 (63.7)	
No	150(36.1)	265 (63.9)		80 (19.3)	335 (80.7)	
Father’s education			0.612			6.736
Primary school or below	34 (44.2)	43 (55.8)		24 (24.7)	73 (75.3)	
Middle school	118 (41.5)	166 (58.5)		73 (23.0)	245 (77.0)	
High school	87 (43.1)	115 (56.9)		34 (23.8)	109 (76.2)	
Vocational school or above	31 (38.8)	49 (61.2)		26 (37.7)	43 (62.3)	
Mother’s education			1.941			1.449
Primary school or below	74 (46.0)	87 (54.0)		34 (21.8)	122 (78.2)	
Middle school	107 (41.2)	153 (58.8)		67 (26.6)	185 (73.4)	
High school	61 (38.6)	97 (61.4)		41 (26.5)	114 (73.5)	
Vocational school or above	28 (43.8)	36 (56.2)		15 (23.4)	49 (76.6)	
Parental marital status			1.374			8.458 ^1^
Married	256 (42.5)	346 (57.5)		146 (24.6)	448 (75.4)	
Divorced/Separated	9 (37.5)	15 (62.5)		9 (52.9)	8 (47.1)	
Other	5 (29.4)	12 (70.6)		2 (12.5)	14 (87.5)	
Age ($\bar{x}$ ± s, years)	13.22 ± 0.91	13.17 ± 0.81	0.687	13.04 ± 0.86	13.04 ± 0.76	0.087
PA exposure ($\bar{x}$ ± s, min/day)	129.22 ± 131.10	74.13 ± 80.09	6.136 ^3^	88.85 ± 81.27	49.45 ± 37.18	5.799 ^3^
Screen time ($\bar{x}$ ± s, h/day)	3.29 ± 2.79	2.75 ± 2.35	2.584 ^2^	2.82 ± 2.32	2.36 ± 1.99	2.387 ^1^
Study duration ($\bar{x}$ ± s, h/day)	8.32 ± 3.28	8.00 ± 2.86	1.267	8.72 ± 3.03	8.94 ± 2.40	0.807
Sleep duration ($\bar{x}$ ± s, h/day)	8.06 ± 1.19	8.00 ± 1.03	0.706	7.95 ± 1.02	7.88 ± 1.01	0.776

* Categorical variables were tested by Pearson chi-square tests, and continuous variables were tested by independent-sample *t*-tests; ^1^
*p* < 0.05; ^2^
*p* < 0.01; ^3^
*p* < 0.001; ^4^ including overweight and at risk for overweight.

**Table 2 ijerph-16-02359-t002:** Comparison of physical activity (PA)-related behaviors in physical activity-related injuries (PARI) and non-PARI group among different genders.

PA-Related Behaviors	Boys (*n* = 643)			Girls (*n* = 627)		
	PARI (*n* = 270) *n* (%)	Non-PARI (*n* = 373) *n* (%)	*χ* ^2^	PARI (*n* = 157) *n* (%)	Non-PARI (*n* = 470) *n* (%)	*χ* ^2^
Doing warm-up			0.670			1.066
Never	32 (43.8)	41 (56.2)		17 (28.8)	42 (71.2)	
Seldom	72 (39.8)	109 (60.2)		37 (23.7)	119 (76.3)	
Sometimes	93 (41.9)	129 (58.1)		61 (26.3)	171 (73.7)	
Often	73 (43.7)	94 (56.3)		42 (23.3)	138 (76.7)	
Doing cool-down			3.773			6.395
Never	54 (37.5)	90 (62.5)		32 (26.2)	90 (73.8)	
Seldom	119 (40.6)	174 (59.4)		56 (20.4)	219 (79.6)	
Sometimes	67 (46.2)	78 (53.8)		45 (29.4)	108 (70.6)	
Often	30 (49.2)	31 (50.8)		24 (31.2)	53 (68.8)	
Using sunscreen			5.527			10.625 ^1^
Never	169 (39.4)	260 (60.6)		54 (20.8)	205 (79.2)	
Seldom	73 (49.7)	74 (50.3)		49 (23.3)	161 (76.7)	
Sometimes	18 (38.3)	29 (61.7)		36 (32.1)	76 (67.9)	
Often	10 (50.0)	10 (50.0)		18 (39.1)	28 (60.9)	
Drinking regularly			7.041			1.982
Never	21 (34.4)	40 (65.6)		11 (19.6)	45 (80.4)	
Seldom	49 (39.8)	74 (60.2)		37 (24.2)	116 (75.8)	
Sometimes	82 (38.1)	133 (61.9)		56 (24.3)	174 (75.7)	
Often	118 (48.4)	126 (51.6)		53 (28.3)	135 (71.8)	
Wearing suitable shoes			4.549			6.987
Never	38 (33.9)	74 (66.1)		15 (14.9)	86 (85.1)	
Seldom	59 (40.4)	87 (59.6)		37 (25.5)	108 (74.5)	
Sometimes	67 (45.6)	80 (54.4)		45 (26.8)	123 (73.2)	
Often	106 (44.5)	132 (55.5)		60 (28.2)	153 (71.8)	
Wearing suitable clothing			2.100			8.794 ^1^
Never	56 (37.3)	94 (62.7)		23 (16.9)	113 (83.1)	
Seldom	94 (43.1)	124 (56.9)		63 (30.7)	142 (69.3)	
Sometimes	60 (42.0)	83 (58.0)		35 (23.2)	116 (76.8)	
Often	60 (45.5)	72 (54.5)		36 (26.7)	99 (73.3)	
Using protective equipment			3.693			2.150
Never	176 (39.6)	268 (60.4)		103 (24.8)	312 (75.2)	
Seldom	67 (45.9)	79 (54.1)		36 (22.9)	121 (77.1)	
Sometimes	16 (50.0)	16 (50.0)		14 (33.3)	28 (66.7)	
Often	11 (52.4)	10 (47.6)		4 (30.8)	9 (69.2)	
Exercising on a wet floor			20.098 ^3^			0.650
Never	113 (34.6)	214 (65.4)		84 (23.9)	268 (76.1)	
Seldom	101 (45.9)	119 (54.1)		57 (26.9)	155 (73.1)	
Sometimes	42 (56.0)	33 (44.0)		14 (25.5)	41 (74.5)	
Often	14 (66.7)	7 (33.3)		2 (25.0)	6 (75.0)	
Exercising on an uneven floor			11.268 ^1^			3.309
Never	106 (35.9)	189 (64.1)		75 (24.0)	237 (76.0)	
Seldom	95 (44.6)	118 (55.4)		59 (28.6)	147 (71.4)	
Sometimes	49 (48.0)	53 (52.0)		20 (23.0)	67 (77.0)	
Often	20 (60.6)	13 (39.4)		3 (13.6)	19 (86.4)	
Exercising in insufficient light			5.464			6.379
Never	72 (37.9)	118 (62.1)		51 (21.9)	182 (78.1)	
Seldom	106 (40.0)	159 (60.0)		60 (23.5)	195 (76.5)	
Sometimes	74 (49.0)	77 (51.0)		41 (33.3)	82 (66.7)	
Often	18 (48.6)	19 (51.4)		5 (31.3)	11 (68.8)	
Exercising in extreme hot weather			9.236 ^1^			9.902 ^1^
Never	69 (40.1)	103 (59.9)		35 (17.9)	160 (82.1)	
Seldom	111 (39.1)	173 (60.9)		76 (28.4)	192 (71.6)	
Sometimes	54 (42.5)	73 (57.5)		34 (25.6)	99 (74.4)	
Often	36 (60.0)	24 (40.0)		12 (38.7)	19 (61.3)	
Exercising in extreme cold weather			16.696 ^2^			11.981 ^2^
Never	42 (39.6)	64 (60.4)		21 (17.2)	101 (82.8)	
Seldom	75 (34.4)	143 (65.6)		64 (25.9)	183 (74.1)	
Sometimes	94 (43.1)	124 (56.9)		46 (23.8)	147 (76.2)	
Often	59 (58.4)	42 (41.6)		26 (40.0)	39 (60.0)	
Exercising with illness			7.836			7.139
Never	139 (39.6)	212 (60.4)		71 (20.8)	270 (79.2)	
Seldom	83 (41.9)	115 (58.1)		55 (29.7)	130 (70.3)	
Sometimes	45 (54.9)	37 (45.1)		26 (31.0)	58 (69.0)	
Often	3 (25.0)	9 (75.0)		5 (29.4)	12 (70.6)	
Wearing glasses			2.285			12.088 ^2^
Never	208 (42.8)	278 (57.2)		112 (23.3)	368 (76.7)	
Seldom	21 (46.7)	24 (53.3)		23 (45.1)	28 (54.9)	
Sometimes	14 (41.2)	20 (58.8)		7 (25.9)	20 (74.1)	
Often	27 (34.6)	51 (65.4)		15 (21.7)	54 (78.3)	
Wearing accessories			1.680			5.830
Never	104 (40.8)	151 (59.2)		40 (20.2)	158 (79.8)	
Seldom	94 (40.3)	139 (59.7)		63 (24.8)	191 (75.2)	
Sometimes	44 (46.3)	51 (53.7)		37 (31.9)	79 (68.1)	
Often	28 (46.7)	32 (53.3)		17 (28.8)	42 (71.2)	
Looking over the PA surroundings			5.110			1.286
Never	29 (31.5)	63 (68.5)		20 (21.3)	74 (78.7)	
Seldom	134 (43.4)	175 (56.6)		89 (24.9)	269 (75.1)	
Sometimes	73 (43.2)	96 (56.8)		36 (27.1)	97 (72.9)	
Often	34 (46.6)	39 (53.4)		12 (28.6)	30 (71.4)	
Concerned about physical status			4.816			5.935
Never	123 (38.7)	195 (61.3)		76 (22.9)	256 (77.1)	
Seldom	108 (43.4)	141 (56.6)		62 (25.5)	181 (74.5)	
Sometimes	32 (53.3)	28 (46.7)		18 (39.1)	28 (60.9)	
Often	7 (43.8)	9 (56.2)		1 (16.7)	5 (83.3)	

^1^*p* < 0.05; ^2^
*p* < 0.01; ^3^
*p* < 0.001.

**Table 3 ijerph-16-02359-t003:** Comparison of Adolescent Risk-taking Questionnaire–Risk Behavior Scale (ARQ-RB) scores in physical activity-related injuries (PARI) and non-PARI group among different genders.

ARQ-RB Factors	Boys (*n* = 643)			Girls (*n* = 627)		
	PARI (*n* = 270)	Non-PARI (*n* = 373)	*t*	PARI (*n* = 157)	Non-PARI (*n* = 470)	*t*
Thrill-seeking behavior	7.51 ± 2.61	6.94 ± 2.44	2.813 ^2^	7.54 ± 2.20	6.83 ±1.84	3.599 ^3^
Rebellious behavior	7.58 ± 2.40	6.95 ± 1.61	3.743 ^3^	7.15 ± 1.85	6.91 ± 1.63	1.529
Reckless behavior	2.11 ± 0.55	2.07 ± 0.34	1.158	2.09 ± 0.43	2.08 ± 0.39	0.168
Anti-social behavior	6.00 ± 2.38	5.49 ± 1.89	2.911 ^2^	5.73 ± 2.72	5.48 ± 1.86	1.244
Total	23.20 ± 5.64	21.45 ± 4.24	4.306 ^3^	22.50 ± 4.70	21.30 ± 3.84	2.879 ^2^

^1^*p* < 0.05; ^2^
*p* < 0.01; ^3^
*p* < 0.001.

**Table 4 ijerph-16-02359-t004:** Multivariate logistic regression to estimate risks factors for physical activity-related injuries (PARI) among boys.

Variables	Partial Regression Coefficient (*β*)	Standard Error (S.E.)	Odds Ratios (OR)	95% Confidence Interval (CI)	*p*-Value
Living in a school dormitory					
No			1 (ref.)		
Yes	0.619	0.179	1.858	1.309–2.637	0.001
Sports team member					
No			1 (ref.)		
Yes	0.397	0.181	1.487	1.042–2.121	0.029
Wet floor					
Never			1 (ref.)		
Seldom	0.614	0.190	1.847	1.274–2.679	0.001
Sometimes	0.741	0.285	2.097	1.200–3.666	0.009
Often	1.103	0.502	3.013	1.127–8.052	0.028
Rebellious behavior	0.110	0.047	1.116	1.017–1.224	0.020
PA exposure (min/day)	0.007	0.001	1.007	1.005–1.009	<0.001

**Table 5 ijerph-16-02359-t005:** Multivariate logistic regression to estimate risks factors for physical activity-related injuries (PARI) among girls.

Variables	Partial Regression Coefficient (*β*)	Standard Error (S.E.)	Odds Ratios (OR)	95% Confidence Interval (CI)	*p*-Value
Sports team member					
No			1 (ref.)		
Yes	0.905	0.202	2.471	1.662–3.673	<0.001
Parental marital status					
Married			1 (ref.)		
Divorced or separated	1.086	0.531	2.962	1.046–8.385	0.041
Other	1.281	0.870	0.278	0.050–1.528	0.141
PA exposure (min/day)	0.013	0.002	1.013	1.009–1.016	<0.001

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
