# Peer review of "Gender-Specific Risk Factors of Physical Activity-Related Injuries among Middle School Students in Southern China"

_ijerph, 2019, doi:10.3390/ijerph16132359_

Round 1
Reviewer 1 Report
Gender-Specific Risk Factors of Physical Activity-Related Injuries Among Middle School Students in Southern China
The authors address an interesting and current theme, however, they did not consider the most important variables for the selected sample. Exposure to risk, type of PA, biological maturation of the sample and parental education were not evaluated.
The methodology presents important gaps in relation to the chosen instruments.
Authors also should consider to improve the discussion section by including other important references to discuss their results.
English revision is needed.
Definition of the sample age is important.
There is not enough information about some of the instruments used. Socio-demographic questionnaire, individual behavior towards PA and PARI experiences need to be carefully explained and more information needs to be provided.
Are these instruments validated for the studied population in a separated study? Provide more information.
The questions made about individual behaviors towards PA have more than one variable. How could you understand and adress the variable to what where the students responding?
In the procedures the author´s only talk about one self-reported questionnaire when they presented 4 questionnaires in the methodology.
What´s the explanation for including variables like single child, nearsightedness, losing weight.
There is no information about the PA that the students were engaged. Exposure time was not calculated. This is is one of the most important risk factors. Also sex is highly related to the type of PA engaged.
Would be better if it was blinded for the investigators who helped the kids to fill the questionnaires.
Why parental instruction was not determined?
The authors don´t refer the age of the students but by reading we understand that this sample is a maturing sample. Since you are comparing boys and girls it was critical that biological variables were considered. You might have been comparing students that are not in the same circunstances considering the biological characteristics. Biological maturation is an important factor, since some studies are already identifying the injury profile for boys and for girls before, during and after peak height velocity. As well being late maturer, on time and early maturre is relevant and each category brings different injuries risks.
The reasons presented for the explanation of the higher risk of PARI is not exclusive of sports teams. Athletes of individual sports present the same issues.
The reasoning for the higher injury rate in sedentary students has more things to it.
In general, authors should consider to improve the discussion section by including other important references with which discuss their results.
Author Response
Point 1: The authors address an interesting and current theme, however, they did not consider the most important variables for the selected sample. Exposure to risk, type of PA, biological maturation of the sample and parental education were not evaluated.
Response 1: Thank you so much for your comments. There are two databases in our study, and we have been used only one of them to write this article. However, considering your valuable comments and suggestions, we integrate the two databases for further analysis in the revision process. In the revised manuscript, we have added parental education, PA exposure time, and further information to explain the biological maturation of the study sample. Please refers to corresponding changes in Table 1 in the Materials and Methods and Results sections.
Point 2: The methodology presents important gaps in relation to the chosen instruments.
Response 2: We have added more details on the instrument used in the study. We have added the information about how the PA exposure time was collected as well. Please refer to corresponding changes in the Materials and Methods section.
Point 3: Authors also should consider to improve the discussion section by including other important references to discuss their results.
Response 3: We have revised and included more relevant references to improve our discussion. Please refer to corresponding changes in the Discussion section.
Point 4: English revision is needed.
Response 4: The revised manuscript has undergone English language editing by MDPI (english-edited-10304).
Point 5: Definition of the sample age is important.
Response 5: We have added more information about the age of our study sample. Please refer to corresponding changes in Study Sample in the Materials and Methods section and in Results section.
Point 6: There is not enough information about some of the instruments used. Socio-demographic questionnaire, individual behavior towards PA and PARI experiences need to be carefully explained and more information needs to be provided.
Response 6: We have added more information about the instrument used in the present study. Please refer to corresponding changes in the Materials and Methods section.
Point 7: Are these instruments validated for the studied population in a separated study? Provide more information.
Response 7: We conducted a pilot study in a class of students to validate the reliability of the instruments. We have added more information about it. Please refer to corresponding changes in the Materials and Methods section.
Point 8: The questions made about individual behaviors towards PA have more than one variable. How could you understand and adress the variable to what where the students responding?
Response 8: Previous study revealed that different individual exercise behaviors would affect the occurrence of PARI. Therefore, the questions consisting of 17 items were used to ask about individual behaviors before, during, and after PA participation. For each item, students were asked to endorse one of four responses: would never do, would hardly ever do, would do sometimes, and would do often. We understand and address the variable according to their response.
Point 9: In the procedures the author´s only talk about one self-reported questionnaire when they presented 4 questionnaires in the methodology.
Response 9: In fact, the structured self-reported questionnaires were comprised of five sections, i.e. socio-demographics, PA exposure time (adding in the revision version), individual exercise behaviors, risk-taking behaviors, and PARI occurrences in the past 12 months. We have made more clarification in the Data Collection and Procedures. Please refer to corresponding changes in the Materials and Methods section.
Point 10: What´s the explanation for including variables like single child, nearsightedness, losing weight.
Response 10: Earlier reports indicated that these variables might influence the occurrence of PARI in other populations. For instance, previous study revealed that students with nearsightedness had low involvement in PA and thus decreased their exposure risk for injury. To further find the association of these variables with PARI among middle school students, we included them in the current study.
Point 11: There is no information about the PA that the students were engaged. Exposure time was not calculated. This is one of the most important risk factors. Also sex is highly related to the type of PA engaged.
Response 11: We totally agree with your comments. In the revised manuscript, we have added information about how the PA exposure time was collected, and analyzed by different groups (eg, boys vs. girls, boarders vs. day-students, etc). Please refer to corresponding changes in the Results and Discussion sections.
Point 12: Would be better if it was blinded for the investigators who helped the kids to fill the questionnaires.
Response 12: Of course, it might be better if it was blinded for the investigators. However, it was difficult for us to do so, especially when our participants were junior middle school students. In the study, the roles of our trained personnel were to orally explain the purpose and meaning of the study and to answer any questions that arose by the study sample during the session, aiming to guarantee the accuracy of the questionnaires. For instance, under the guidelines of our investigators, PARI episode reported and counted by students would be in line with the definition and criteria, which would be help to validate the outcome PARI measure.
Point 13: Why parental instruction was not determined?
Response 13: Prior to approaching potential subjects, explanatory statements and consent forms were distributed to all eligible students’ parents, aiming to give them the details and instruction of our study and receive more support and consent from them.
Point 14: The authors don´t refer the age of the students but by reading we understand that this sample is a maturing sample. Since you are comparing boys and girls it was critical that biological variables were considered. You might have been comparing students that are not in the same circunstances considering the biological characteristics. Biological maturation is an important factor, since some studies are already identifying the injury profile for boys and for girls before, during and after peak height velocity. As well being late maturer, on time and early maturre is relevant and each category brings different injuries risks.
Response 14: We totally agree with your comment that biological maturation is an important factor for PA engagement and PARI occurrence. In this study, our eligible participants were in grades 7th and 8th from six middle schools, who aged from 12 to 16 years old, with a mean age of 13.12 (SD: 0.83). All of them were pre-maturing adolescents, and we compared the whole study sample in the similar biological characteristics (mean age of boys: 13.14±0.71; girls: 13.12±0.69; P > 0.05).
Point 15: The reasons presented for the explanation of the higher risk of PARI is not exclusive of sports teams. Athletes of individual sports present the same issues.
Response 15: In the present study, we found that sports team members were more likely to sustain PARI in both boys and girls, which was in line with previous reports. Of course, athletes of individual sports were also at a higher risk of PARI due to their higher PA levels. However, in this study, sports team member did not refer to professional athletes, but those who took part in school sports teams out of individual interest in their leisure time.
Point 16: The reasoning for the higher injury rate in sedentary students has more things to it.
Response 16: We totally agree with this comment. Those sedentary students usually have more electronic screen watching time. This would lead to the lower rate of PA participation for them. Just as the findings from other reports, the lower PA level they engage, the higher risk of physical and mental problems they have, and the higher risk of PARI occurrence they experience.
Point 17: In general, authors should consider to improve the discussion section by including other important references with which discuss their results.
Response 17: We have included more important and related references to improve our discussion. Please refer to corresponding changes in the Discussion section.
Reviewer 2 Report
It does not refer, in sufficient quantity and in an adequate manner, summarizing and commenting, as appropriate, on other research or work carried out in the field of the subject matter addressed. This aspect of the work should be improved with a thematic and temporary organization that considers appropriate authority figures.
It does not provide recommendations, discuss, reflect or call for action on the results obtained, nor is there any internal consistency between the discussion and the conclusions.
There is no certain adjustment between the problem and the objectives. Nor relevance and sufficiency
Author Response
Point 1: It does not refer, in sufficient quantity and in an adequate manner, summarizing and commenting, as appropriate, on other research or work carried out in the field of the subject matter addressed. This aspect of the work should be improved with a thematic and temporary organization that considers appropriate authority figures.
Response 1: Thank you so much for your comments. We have made some adjustment for this problem. Please refers to corresponding changes in the revised manuscript.
Point 2: It does not provide recommendations, discuss, reflect or call for action on the results obtained, nor is there any internal consistency between the discussion and the conclusions.
Response 2: We have made some adjustment for this problem. Please refers to corresponding changes in the revised manuscript.
Point 3: There is no certain adjustment between the problem and the objectives. Nor relevance and sufficiency
Response 3: We have made some adjustment for this problem. Please refers to corresponding changes in the revised manuscript.
Reviewer 3 Report
In their manuscript titled: “Gender-specific risk factors of physical activity-related injuries among middle school students in Southern China”, the authors Tang et al. describe the result of their cross-sectional survey-design study to determine sport-related injuries among middle school students. Responses from boys and girls are analysed and reported on separately, which is a great way to sort out potential interaction effects between gender and other factors. There are, however, some study concerns and limitations that need to be addressed.
1. The study does not take exposure into account. Therefore, the ‘injury risk factors’ could in fact be factors associated with increased sports participation. For example, living in the school dormitory was found to be associated with sports injury, particularly among boys. Rather than the explanation of potential junk food consumed by boarders, as suggested in the discussion, I think boarders might participate in much more sport/physical activity. Without accounting for exposure, there is no way of knowing if any of the results are actually factors associated with injury, or factors associated with physical activity. Exposure need to detail not only sports participation, but duration, intensity, team membership and outdoor activities.
2. The study is cross sectional in design. Therefore, risk factors and history of injury are captured at the same time-point. Causality in the association cannot be established; furthermore, time sequences cannot be established. For example, did those with injury have more screen time because they were incapacitated due to their injury? Rather than screen time leading to injuries?
3. The study needs a conceptual framework on which the survey questions should be based. Which factors should be explored, and why? Currently, many of the questions are not effective, or it is unclear why they were included. An example of an ineffective question is the question regarding sun screen. This could be considered a proxy for participation in outdoor sports (hence exposure) and for proactive healthy behaviour (preventing sunburn). What was the rationale for including questions on the parents’ marital status?
4. There is no explanation of how potential participants (and parents) were approached, participation rates, survey completion rates, and potential response bias. Dates for the study period also need to be provided.
5. A sample size calculation, based on expected participation rates and required level of statistical power, would be useful.
6. How were the schools selected, and what was the rational regarding which schools were selected?
Author Response
Response to Reviewer 3 Comments
In their manuscript titled: “Gender-specific risk factors of physical activity-related injuries among middle school students in Southern China”, the authors Tang et al. describe the result of their cross-sectional survey-design study to determine sport-related injuries among middle school students. Responses from boys and girls are analysed and reported on separately, which is a great way to sort out potential interaction effects between gender and other factors. There are, however, some study concerns and limitations that need to be addressed.
Point 1: The study does not take exposure into account. Therefore, the ‘injury risk factors’ could in fact be factors associated with increased sports participation. For example, living in the school dormitory was found to be associated with sports injury, particularly among boys. Rather than the explanation of potential junk food consumed by boarders, as suggested in the discussion, I think boarders might participate in much more sport/physical activity. Without accounting for exposure, there is no way of knowing if any of the results are actually factors associated with injury, or factors associated with physical activity. Exposure need to detail not only sports participation, but duration, intensity, team membership and outdoor activities.
Response 1: Thank you so much for your comments. We totally agree with this comment that PA exposure time plays an important role in the development and occurrence of PARI. There are two databases in our study, and we have been used only one of them to write this article. However, considering your valuable comments and suggestions, we integrate the two databases for further analysis in the revision process. In the revised manuscript, we have added PA exposure time for further analysis and clarified how it was collected. Please refers to corresponding changes in Table 1 in the Results and the Materials and Methods sections.
Point 2: The study is cross sectional in design. Therefore, risk factors and history of injury are captured at the same time-point. Causality in the association cannot be established; furthermore, time sequences cannot be established. For example, did those with injury have more screen time because they were incapacitated due to their injury? Rather than screen time leading to injuries?
Response 2: We totally agree with this comment. The nature of cross-sectional study prevents us from causal and temporal inferences between PARI occurrences and the potential risk factors. It is for example plausible that those injured students would rate their level of PA higher if they were not injured due to the adverse influence on PA level. Likewise, it is also plausible that those students who take part in PA more often might remember their PARI experience more clearly since their injuries might have a bigger impact on these physically active students. Therefore, this study should be viewed as exploratory and preliminary due to the absence of longitudinal data. We are conducting a prospective study to avoid this problem and to confirm those cause-effect relationships found in the present study.
Point 3: The study needs a conceptual framework on which the survey questions should be based. Which factors should be explored, and why? Currently, many of the questions are not effective, or it is unclear why they were included. An example of an ineffective question is the question regarding sun screen. This could be considered a proxy for participation in outdoor sports (hence exposure) and for proactive healthy behaviour (preventing sunburn). What was the rationale for including questions on the parents’ marital status?
Response 3: Most of the variables investigated in the current study were based on the previous study, and we had added more references to clarify these survey questions. Please refers to corresponding changes in the revised manuscript. Using sun screen have been identified as an individual behavior when students participated in PA during the hottest hours, though just as you mentioned that using sun proof might be considered a proxy for participation in outdoor activities and for proactive healthy behavior. Furthermore, we included parental marital status into the current study since previous reports revealed that family context especially single-parent families contributed to the increased risk of getting injured, and our study had obtained the rationale that girls’ students living in families whose parents were divorce or separation had more likely to suffer from PARI. For more details, please refer to corresponding changes in the Discussion section.
Point 4: There is no explanation of how potential participants (and parents) were approached, participation rates, survey completion rates, and potential response bias. Dates for the study period also need to be provided.
Response 4: We have made clarification about this problem. Please refer to corresponding changes in the Procedures section.
Point 5: A sample size calculation, based on expected participation rates and required level of statistical power, would be useful.
Response 5: We had calculated the sample size using the following formula before the implement of this study:
n=(〖u_α〗^2 p_0 (1-p_0))/〖(p-p_0)〗^2
where α=0.05, p_0=0.25, (p-p_0)=0.1p_0, so the number of sample size was 1152 accordingly. Considering the existence of unanswered and invalid questionnaires, we increased the sample size by 20%, so the final sample size of the cross-sectional survey was 1267. Thus, the sample size of the current study met the basic requirement.
Point 6: How were the schools selected, and what was the rational regarding which schools were selected?
Response 6: We have made clarification about this problem. Please refer to corresponding changes in the Study Sample section.
Round 2
Reviewer 1 Report
Dear Authors
Please see my comments below:
The questions made about individual behaviors towards PA have more than one variable. How could you understand and adress the variable to what where the students responding? After the revision i still don’t understand how the information was considered since the questions had more than one variable.
The multivariate logistic regression tables were changed. What´s the reason?
The odds ratios of the PA exposure variable are really low for boys and girls.
Biological maturation wasn’t determinated.
Nevertheless the changes made were substancial and the quality of the article really improved.
Best regards
Author Response
Point1: The questions made about individual behaviors towards PA have more than one variable. How could you understand and adress the variable to what where the students responding? After the revision i still don’t understand how the information was considered since the questions had more than one variable.
Response1: Thank you so much for your comments. Maybe I did not make clear clarification about the individual exercise behaviors, so I make further revision of this part. The scale regarding individual behaviors used in our study was display as follow. For instance, (a) “Did you do warm-up exercises before PA participation?”, (b) “Did you do cool-down exercises after PA participation?”, (c) “Did you use sunscreen during PA participation?”, (d) “Did you drink water regularly during PA participation?”, (e) “Did you wear suitable shoes during PA participation?”, (f) “Did you wear suitable clothing during PA participation?”, (g) “Did you wear protective equipment during PA participation?”, (h) “Did you undertake PA on a wet floor?”, (i) “Did you undertake PA on an uneven floor?”, (j) “Did you undertake PA in insufficient lights?”, (k) “Did you undertake PA in extreme hot hours?”, (l) “Did you undertake PA in extreme cold hours?”, (m) “Did you undertake PA with illness?”, (n) “Did you wear glasses during PA participation?”, (o) “Did you wear accessories during PA participation?”, (p) “Did you look over the surroundings before PA participation?”, (q) “Did you pay attention to your physical status during PA participation?”. For each item, students were asked to endorse one of these four responses: would never do, would hardly ever do, would do sometimes, and would do often. You can check the following attachment (Chinese and English version) to learn more about it.
Point2: The multivariate logistic regression tables were changed. What´s the reason?
Response2: Considering your valuable comments and suggestions on PA exposure time, we have added this variable for further analysis in the revised version (Round 1). PA exposure time was found to be a significant variable tested by independent-sample t-tests and we therefore included it with other significant variables tested by t-tests or chi-square tests to perform a multivariate logistic regression analysis. Thus, the results of significant variables kept in the final multivariate logistic regression model changed.
Point3: The odds ratios of the PA exposure variable are really low for boys and girls.
Response3: Yes. The duration of PA participation was significantly associated with PARI occurrence for both boys and girls with the low odds ratios. However, if we consider it deeply, the ORs may contribute significantly to PARI occurrence. Take boys for example, students with longer PA exposure time (mins) is 1.007 times of suffering PARI. That is, additional 1 minute of PA exposure increases the likelihood of PARI by 0.007 times and 100 minutes of PA exposure increases the likelihood of PARI by 0.7 times. The similar results of PA exposure variable could be found in other study for university students (Physical activity-related injuries among university students: a multicentre cross-sectional study in China).
Point4: Biological maturation wasn’t determinated.
Response4: To be honest, we did a literature search in Medline when we designed the survey questionnaire. We found that biological maturation was a significant factor for PARI, especially for young athletes. We therefore included this question in our first version of pilot questionnaire. In this study, we did not determine the biological maturation due to the following reasons: 1) previous studies revealed that biological maturation was an important factor of the injury profile for both boys and girls before, during and after peak height velocity. Although middle school students have a certain understanding of their own growth and development, especially the obvious changes brought about by adolescence. However, this question may be sensitive, particularly in Chinese culture; 2) we conducted a pilot study, in which a class of students were asked to report the detail of this question. But we found that they did not answer this question completely or even did not answer it. We therefore decided to delete this question on the current version to avoid the incompletion; 3) we found that the distribution of age between PARI and non-PARI groups was similar in both boys and girls (please check the additional data of age in Table 1 and the following attachment). Nevertheless, we agree with your valuable comments about this question and the inclusion of biological characteristics may be more helpful to explore its relationship with PARI occurrences among general populations. We will take it into consideration in our further study, especially our study participants aged 12 to 16. Thank you so much for it.

Reviewer 3 Report
Study titled: Gender-specific risk factors of physical activity-related injuries among middle school students in Southern China
The authors have addressed some of the concerns raised in the review process, and the resulting manuscript is improved. There are, however, some issues that remain unaddressed.
There is still no rationale as to the variables that are collected and analysed. For example, the study could be aiming to identify those at risk of injury, to be able to target preventive measures to this group. Alternatively, the aim of the study could be to identify causal factors leading to injury, so that these factors can be modified to prevent the injuries from occurring. The latter would require proving causality. Some of the variable collected, such as marital status of the parents, are suitable for neither of these aims.
Including physical activity levels in the study is a substantial improvement. However, for the interpretation of Table 1, all results should be reported as rates per physical activity.
Discussion, lined 263-264: where are the results regarding screen and study time of boarders vs. non-boarders? In presenting these, please also add physical activity of boarders vs. non-boarders.
Minor:
Methods, page 3 lines 98: ‘elevate’ should be ‘evaluate’?
Methods, page 3 line 128: the categories should be re-named. I don’t think overeating can be categorised as anti-social behaviour.
Methods, page 4 line 150: ‘serious incompletion’ should be rephrased. Ideally, the threshold can be quantified – how many items need to be completed for the case to be included.
Author Response
Point1: There is still no rationale as to the variables that are collected and analysed. For example, the study could be aiming to identify those at risk of injury, to be able to target preventive measures to this group. Alternatively, the aim of the study could be to identify causal factors leading to injury, so that these factors can be modified to prevent the injuries from occurring. The latter would require proving causality. Some of the variable collected, such as marital status of the parents, are suitable for neither of these aims.
Response1: Thank you so much for your comments. PARI is a health issue that involves multifaceted factors in terms of individual, environmental, social, etc. In this study, we included most of the investigated variables mainly on the basis of the previous reports. Just as the statement in the last paragraph of the Introduction section, the major aim of this cross-sectional study was to preliminarily explore significant multidimensional factors associated with PARI occurrence for different genders that could provide a basis for us to take targeted gender-specific prevention and intervention measures to reduce the occurrence of PARI. We agree with you that this cross-sectional study cannot provide case-effect relationship due to the absence of longitudinal data. According to our proposal, this study serves as a baseline of a mixed study supported by the General Administration of Sport of China (Grant No. 2017B045) to develop and implement injury-prevention strategies for those students who are high-risk, vulnerable group. In addition, our team is carrying out a prospective design to identify casual factors leading to PARI. According to the previous study, family context was a potential contributor to injury. As a part of family context, parental marital status (especially the parents were divorced or separated) was associated with PARI occurrence. In this study, we found that female students living in families where are divorced or separated had more possibility to suffer from PARI with the odds ratio of 2.962. Admittedly, we are unable to modify this variable to prevent the injuries from occurring, but we can therefore place great emphasis on those high-risk, vulnerable children in single-parent families and take focused and effective actions to prevent them from PARI and maximize the benefits of PA.
Point2: Including physical activity levels in the study is a substantial improvement. However, for the interpretation of Table 1, all results should be reported as rates per physical activity.
Response2: It may be difficult or inappropriate for us to report the rates per physical activity. In this study, we used CLASS-C to collect the frequency and duration of 20 types of MVPAs (including Moderate-intensity PA and Vigorous-intensity PA) and found that not every student would participate in every kind of PA. In fact, basketball, walking, cycling, and running were the most commonly PA that students participated. The participation rate of the PA item like martial art, volleyball, jumping, and tennis were rather low. Therefore, we think it may be more suitable to report the total duration of different kinds of PA to reflect the PA participation per student.
Point3: Discussion, lined 263-264: where are the results regarding screen and study time of boarders vs. non-boarders? In presenting these, please also add physical activity of boarders vs. non-boarders.
Response3: We have added the related results accordingly.
Minor:
Point4: Methods, page 3 lines 98: ‘elevate’ should be ‘evaluate’?
Response4: We have revised it accordingly.
Point5: Methods, page 3 line 128: the categories should be re-named. I don’t think overeating can be categorised as anti-social behaviour.
Response5: The categories of risk-taking behaviors were divided according to the previous studies. The Chinese version of Adolescent Risk-Taking (ARQ-RB, including 17 items that were divided into four factors) was revised from the original version (including 22 items that were divided into four factors). In the both versions, overeating was categorized as anti-social behavior. In addition to the Chinese version of ARQ-RB, you can review the following original paper to get more details: Gullone E, Moore S, Moss S, et al. The Adolescent Risk-Taking Questionnaire: Development and Psychometric Evaluation[J]. Journal of Adolescent Research, 2000, 15(2):231-250. I hope this could make you have a good understanding about it.
Point6: Methods, page 4 line 150: ‘serious incompletion’ should be rephrased. Ideally, the threshold can be quantified – how many items need to be completed for the case to be included.
Response6: “Serious incompletion” means the incompletion rate of more than 80% (about 50 items), we have rephrased our statement accordingly.